# Identification of the Xyloglucan Endotransglycosylase/Hydrolase (*XTH*) Gene Family Members Expressed in *Boehmeria nivea* in Response to Cadmium Stress

**DOI:** 10.3390/ijms232416104

**Published:** 2022-12-17

**Authors:** Yu-Shen Ma, Hong-Dong Jie, Long Zhao, Xue-Ying Lv, Xiao-Chun Liu, Yan-Yi Tang, Ying Zhang, Peng-Liang He, Hu-Cheng Xing, Yu-Cheng Jie

**Affiliations:** 1College of Agronomy, Hunan Agricultural University, Changsha 410128, China; 2Hunan Provincial Engineering Research Center for Grass Crop Germplasm Innovation and Utilization, Changsha 410128, China

**Keywords:** *Boehmeria nivea*, *XTH* gene family, genome-wide identification, Cd stress, expression analysis

## Abstract

Xyloglucan endotransglycosylase/hydrolase (XTH) genes play an important role in plant resistance to abiotic stress. However, systematic studies of the response of *Boehmeria nivea* (ramie) *XTH* genes (*BnXTHs*) to cadmium (Cd) stress are lacking. We sought to identify the *BnXTH*-family genes in ramie through bioinformatics analyses and to investigate their responses to Cd stress. We identified 19 members of the *BnXTH* gene family from the ramie genome, referred to as *BnXTH1-19*, among which *BnXTH18* and *BnXTH19* were located on no chromosomes and the remaining genes were unevenly distributed across 11 chromosomes. The 19 members were divided into four groups, Groups I/II/IIIA/IIIB, according to their phylogenetic relationships, and these groups were supported by analyses of intron–exon structure and conserved motif composition. A highly conserved catalytic site (HDEIDFEFLG) was observed in all BnXTH proteins. Additionally, three gene pairs (*BnXTH6*–*BnXTH16*, *BnXTH8*–*BnXTH9*, and *BnXTH17*–*BnXTH18*) were obtained with a fragment and tandem-repeat event analysis of the ramie genome. An analysis of cisregulatory elements revealed that *BnXTH* expression might be regulated by multiple hormones and abiotic and biotic stress responses. In particular, 17 cisregulatory elements related to abiotic and biotic stress responses and 11 cisregulatory elements related to hormone responses were identified. We also found that most *BnXTH* genes responded to Cd stress, and *BnXTH1*, *BnXTH3*, *BnXTH6*, and *BnXTH15* were most likely to contribute to the Cd tolerance of ramie, as evidenced by the substantial increases in expression under Cd treatment. Heterologous expression of *BnXTH1*, *BnXTH6*, and *BnXTH15* significantly enhanced the Cd tolerance of transgenic yeast cells. These results suggest that the *BnXTH* gene family is involved in Cd stress responses, laying a theoretical foundation for functional studies of *BnXTH* genes and the innovative breeding of Cd-tolerant ramie.

## 1. Introduction

Cadmium (Cd) is a nonessential trace metal with high mobility and toxicity [1]. It accumulates in plants through polluted farmland, soil, and water sources and enters the human body through the food chain, thus affecting human health [2]. According to statistics, more than 5 million soils worldwide, mainly in developing and underdeveloped countries (India, Bangladesh, Pakistan, etc.), are polluted with heavy metals [3]. China has the most serious heavy-metal pollution among these countries, with several “Cd rice” incidents [4]. A survey of cultivated land in China showed that about 2.79 × 10^9^ m^2^ of agricultural land, accounting for 20% of the total cultivated land area, is polluted with Cd [5,6]. The southern provinces, such as Hunan, Guizhou, Guangdong, Guangxi, and Fujian, have the most Cd pollution mainly because they are mining areas and smelting-process zones of nonferrous metals; their wastewaters are used for irrigation without treatment, polluting the surrounding cultivated soils [7,8,9]. Therefore, Cd pollution poses a major threat to global food security and human health, necessitating urgent mitigation.

Phytoremediation is a better mitigation measure for Cd pollution, involving Cd enrichment and accumulation in plants from contaminated soils [10]. Ramie is a nonedible perennial plant with rapid growth, a high biomass, a high Cd tolerance and enrichment ability, and a high economic value. Therefore, ramie is an ideal plant resource for remediation of Cd-contaminated soils [11]. Recent evidence has shown that cell walls, especially hemicellulose, may affect the Cd tolerance of ramie, since most Cd is enriched in hemicellulose [12]. However, it is not clear how hemicellulose binds to Cd. Therefore, it is necessary to further explore the role and the potential physiological and molecular mechanisms of action of hemicellulose in Cd accumulation and tolerance.

Hemicelluloses can be divided into four classes: xylans, mannans, β-glucans with mixed linkages, and xyloglucans [13]. Xyloglucans are the most abundant hemicellulose in the primary cell walls of dicotyledons and nonGramineae monocotyledons [14]. Their synthesis requires a glycosidic-bond synthase and various glycosyltransferases [15]. Xyloglucan endoglycosidase/hydrolase (XTH), a key xyloglucan-modifying enzyme belonging to the glycoside hydrolase 16 (GH16) family, catalyzes the cleavage and polymerization of xyloglucan molecules, thereby modifying the cellulose–xyloglucan composite structure of the cell wall [16]. XTH exhibits two catalytic functions: xyloglucan endohydrolase (XEH) activity, which catalyzes hydrolysis of xyloglucan, and xyloglucan endotransglucosylase (XET) activity, which cuts and rejoins xyloglucan chains [17].

*XTH* gene family members have been identified in many species, including *Arabidopsis thaliana* (33 members) [18], *Nicotiana tabacum* (56 members) [19], *Oryza sativa* (29 members) [20], *Medicago truncatula* (44 members) [21], *Brassica rapa* (53 members) [22], *Poplar* spp. (41 members) [23], *Glycine max* (61 members) [24], and *Schima superba* (34 members) [25]. Based on their phylogenetic relationships, *XTH* genes are classified into three major groups: Group I, Group II, and Group III [18]; however, some scholars have further divided Group-III members into groups IIIA and IIIB [20].

Recent research of *XTH* genes has focused on abiotic stress responses, including osmotic, salt, and low-temperature stress responses. Unique members of the *XTH* gene family have also been identified in many species in response to abiotic stress. For example, in *A. thaliana* roots, *AtXTH14*, *AtXTH15*, and *AtXTH31* are downregulated in response to aluminum (Al) stress [26]. In *M. truncatula*, 28 *MtXTH* genes respond to Hg stress, 21 respond to salt stress, and another 21 respond to drought stress [21]. Three homologous genes, *CaXTH1*, *CaXTH2*, and *CaXTH3,* were found to respond to drought, high salt, and low-temperature stress in pepper [27]. The functions of some *XTH* genes have also been studied. For example, *PvXTH9* and *PvXTHb* were associated with Al accumulation in the cell wall of the common bean [28]. Overexpression of *CaXTH3* improves drought and salt tolerance in transgenic tomatoes [29], and overexpression of *AtXTH31* improves flooding-stress tolerance in *Glycine max* [30]. Moreover, *AtXTH19* can improve the freezing tolerance of *A. thaliana* after cold and subzero acclimation [31].

This study identified and analyzed *BnXTH* genes from the ramie genome to reveal the role of *XTH*-family genes in response to Cd stress. We also explored the responses of these genes to Cd stress through yeast expression experiments. The findings of this study provide a theoretical basis for functional studies of *BnXTH*-family genes in ramie.

## 2. Results

### 2.1. Identification of Chromosomal Locations and Physicochemical-Property Analysis of the BnXTH Gene Family

A total of 19 *BnXTH* genes were identified from published ramie genome data. The genes were denoted *BnXTH1-19,* among which *BnXTH6-19* were named based on chromosomal positions (Appendix A and Table 1). As illustrated in Appendix A, *BnXTH18* and *BnXTH19* were located on the Scaffold16 fragment and not on the chromosome, which may have been due to the poor assembly of the ramie genome. The other 17 *BnXTH* genes were unevenly distributed on chromosomes 2, 4, 5, 6, 7, 8, 9, 11, 12, 13, and 14. Chromosomes 1, 3, and 10 had no *XTH* genes, and chromosome 6 contained the largest number of *BnXTH* genes (3; 15.79%), followed by chromosomes 4, 5, and 14, which contained two *BnXTH* genes each. The remaining chromosomes only had one *BnXTH* member each. This study also found that the number of family genes mapped in the chromosomes had no correlation with chromosome length. To further understand the physical and chemical properties of the BnXTH-protein family, we evaluated each protein’s CDS length, amino acid (aa) number, molecular weight (Mw), isoelectric point (PI), grand average of hydropathicity (GRAVY), and aliphatic index. We also predicted the subcellular locations of these proteins (Table 1). These results showed that out of the 19 BnXTH protein sequences, the shortest was BnXTH10, which was encoded by 264 amino acids, while BnXTH3 was the longest, encoded by 395 amino acids. The Mw of the BnXTHs ranged from 30.812 (BnXTH10) to 40.228 kDa (BnXTH4), while the GRAVY ranged from −0.800 (BnXTH12) to −0.223 (BnXTH5). The aliphatic index of the proteins was between 59.44 (BnXTH12) and 73.39 (BnXTH7), and the PI ranged from 4.64 (BnXTH17) to 9.47 (BnXTH9). Moreover, subcellular localization prediction showed that BnXTH13/17 might have been located in extracellular regions, while BnXTH3/7/9/15/18/19 may have been located in the cell wall or cytoplasm, and the remaining BnXTH proteins may have played roles in the cell wall.

### 2.2. Phylogenetic Analysis and Multiple Sequence Alignment of BnXTHs

To better understand the evolutionary relationships between *BnXTHs* and determine their classifications, we used the sequences of the 19 ramie *BnXTH* genes and the AtXTH family protein sequences of *A. thaliana* to generate a phylogenetic tree (Figure 1) using the maximum likelihood (ML) method. We divided the BnXTH-family proteins identified in ramie into four subgroups based on a previous classification of the family: namely, Groups I, II, IIIA, and IIIB. The BnXTHs were mainly clustered in Groups I and II, which had 14 members. Among these members, BnXTH1/2/3/7/10/11/12 belonged to Group I, while BnXTH8/9/13/17/15/18/19 belonged to Group II. The remaining BnXTHs (BnXTH4/5/14/6/16) were included in Groups IIIA and IIIB.

Multiple alignments of the 19 BnXTHs showed that the BnXTH proteins contained a highly conserved catalytic site (HDEIDFEFLG) (Figure 2), with a deviation of one or two amino acids in a few sequences (Figure 2a). Except for the BnXTHs in Group III (BnXTH5/6/16), the active catalytic regions (HDEIDFEFLG) (shown with rectangular purple frames in Figure 2b) of the BnXTHs were adjacent to the N-linked glycosylation site.

### 2.3. Structural Analysis of the Conserved Motifs of BnXTHs

To analyze the gene structures and conserved motifs of *BnXTHs*, we constructed an evolutionary tree using 19 BnXTH proteins (Figure 3a), which were grouped into four subclasses. Structural analysis of the genomic DNA sequence showed that each *BnXTH* had two or three introns (Figure 3b) and that the members in Group I, except for *BnXTH10*, each contained three introns. The Group II members, except for *BnXTH13*, contained two introns each, while the Group IIIA and Group IIIB members had three introns each. Furthermore, MEME analysis showed that 10 conserved motifs were found in the 19 BnXTH protein sequences (Figure 3c). The amino acid sequence that encoded the protein sequences and the SeqLogo of the 10 conserved motifs are shown in Appendix A. Motifs 1, 2, 3, 4, 5, and 6 were abundant in BnXTH proteins, among which motif 2 contained the characteristic active site (HDEIDFEFLG), suggesting that it is the specific motif for the enzymatic reaction of this family of proteins and present in all BnXTHs. We also found that motif 8 seemed unique to Group I, while motif 10 mainly existed in Groups I and II. Thus, these results also support the group-classification results of the phylogenetic tree above.

### 2.4. Gene-Duplication Analysis of BnXTHs

To determine the relationships between *BnXTH* members, we used the MCScanX method for a collinearity analysis. Six genes (*BnXTH6*–*BnXTH16*, *BnXTH8*–*BnXTH9*, and *BnXTH17*–*BnXTH18*) exhibited complex segmental duplication events (Figure 4a), implying that segmentally duplicated genes may have similar functions regulated via the same biological pathways in ramie. To further evaluate the evolution and development of the *BnXTH* family, we compared the collinearity of *XTH* genes between *B. nivea* and four other plants (*O. sativa*, *S. bicolor*, *A. thaliana*, and *P. trichocarpa*) (Figure 4b–e). These results showed that three pairs of homologous genes existed between *B. nivea* and *O. sativa*, while two pairs existed between *B. nivea* and *S. bicolor*. Moreover, 18 pairs of homologous genes were detected between *B. nivea* and *A. thaliana*, and 21 pairs were identified between *B. nivea* and *P. trichocarpa* (Appendix A). These results show that ramie *XTHs* have less homology with monocotyledons than with dicotyledons, and thus, we speculate that these *XTH* genes may be involved in differentiation of dicotyledons.

### 2.5. Ciselement Analysis of the BnXTHs

We used the PlantCARE service to analyze the cisregulatory elements of *BnXTHs* in the upstream sequences (~2000 bp) of their promoters that were associated with response to abiotic and biotic stress, phytohormone signaling, and plant growth and development. We predicted 56 related cisacting elements (Figure 5 and Appendix A), among which 17 were involved in abiotic- and biotic-stress responses. ARE, MYC, STRE, and MBS were the most abundant among the 17 ciselements. Moreover, 11 cisregulatory elements were related to plant hormone responses, including abscisic acid (ABA), gibberellin (GA), salicylic acid (SA), methyl jasmonate (MeJA), and auxin responses. These elements included TGACG motifs, ABREs, TCA elements, TGA elements, and other related ciselements. There were 28 cisregulatory elements related to plant growth and development, among which light-responsive elements, including GT1 motifs, TCCC motifs, GATA motifs, Sp1, and other related elements, were the most abundant.

### 2.6. Expression Analysis of the BnXTH-Family Genes under Cd Treatment

To determine whether *BnXTH*-family genes are involved in Cd stress responses, we treated ramie seedlings with CdCl_2_ and sampled the roots at 0, 3, 6, 9, 12, 24, and 48 h after this treatment. The expression levels of the 19 *BnXTH* genes (Figure 6) were analyzed with RT-qPCR, and these results showed that expression of *BnXTH1/6/15*, especially *BnXTH1*, increased significantly under Cd treatment. The expressions of the *BnXTH1* genes at 6 h and 9 h after the treatment were 5.78 and 6.34 times higher than that at 0 h, respectively. *BnXTH3* was slightly upregulated and had the highest expression at 12 h: 41.22% higher than that at 0 h. Conversely, expression of *BnXTH4/9/14/19* genes had no response to the Cd treatment, while the other *BnXTH* genes exhibited were downregulated under the Cd treatment. In general, the *BnXTH*-family genes responded differently to Cd stress.

### 2.7. Functional Analysis of BnXTHs in Yeast

Since the expression levels of *BnXTH1/3/6/15* genes were upregulated in response to Cd stress, we selected *BnXTH1*, *BnXTH3*, *BnXTH6*, and *BnXTH15* as candidate genes and analyzed their roles in Cd tolerance. The full-length CDSs of *BnXTH1*, *BnXTH3*, *BnXTH6*, and *BnXTH15* were cloned and ligated into yeast expression vector p426 GPD and introduced into Cd-sensitive yeast mutant *Δyap1*. The Cd-tolerance characteristics of transgenic yeast were then analyzed (Figure 7). We found no significant difference in the growth between the *BnXTH1/3/6/15* transgenic yeast strains and the empty vector on the SD-URA medium without Cd. However, the growth of the yeast strains that contained the empty vector was significantly lower compared to that in the *BnXTH1/6/15*-containing yeast strains, particularly the *BnXTH1*-carrying yeast strains, on the medium that contained 75 μM of Cd. These results suggest that *BnXTH1/6/15* genes play important roles in Cd tolerance. However, the growth of the transgenic yeast strain that contained *BnXTH3* was not significant under Cd stress, indicating that *BnXTH3* had no Cd tolerance. These results show that the *BnXTH* gene family is related to the Cd tolerance of ramie, and the three Cd-tolerant genes (*BnXTH1/6/15*) can be used for genetic improvement of Cd tolerance in ramie.

## 3. Discussion

### 3.1. Evolutionary Characteristics of the BnXTH Gene Family in Ramie

A total of 19 *BnXTH* genes were identified in ramie, compared with the 33 family members in *A. thaliana* and 29 in *O. sativa*, suggesting that there are many fewer members of the *BnXTH* gene family in ramie. This may have been due to the incomplete assembly of the genome, resulting in a failure to identify other members of the *BnXTH* gene family or a loss of several *BnXTH* genes in the genome [32]. This result is consistent with those of a previous study that showed that *Brassica oleracea* contains fewer *XTH* family members than those in *Brassica rapa* [22]. These phylogenetic results showed that *BnXTHs* were relatively conserved at the DNA and protein levels and could be divided into four subclasses (Figure 1), similar to those of other plants [33,34]. There were 14 *BnXTH* members in Groups I and II, accounting for 73.68% of all members. When only *BnXTH*-family proteins were used to construct an evolutionary tree, members of Groups I and II clustered together, making it difficult to distinguish them (Figure 3a). This was similar to the findings of previous studies, which reported no significant difference between members of Groups I and II, suggesting that the two can be combined into a larger group: that is, Group I/II [35,36]. Furthermore, a multiple sequence alignment showed that the members of Groups I, II, and IIIB (except for *BnXTH5*) were located near the XET catalytic site (HDEIDFEFLG) and had typical N-glycosylation residues, while Group IIIA members lacked a consensual N-glycosylation site. The same phenomenon was reported in *A. thaliana* [37,38]. Additionally, variation of the conserved catalytic motif (HDEIDFEFLG) was observed in BnXTH proteins, consistently with previous studies involving plants, such as *A. thaliana* [18] and *P. trichocarpa* [14]. Variation in this motif may affect its enzyme catalytic activity, which necessitates further studies of the enzyme activity and other functions of these motif-variant genes.

### 3.2. Evolutionary Analysis of the BnXTH Gene Family

It is speculated that *A. thaliana* experienced genome-wide duplication thrice in the past 250 million years, while *O. sativa* experienced genome-wide duplication about 70 million years ago [39]. As early as 1970, scholars proposed that gene replication is the basic mechanism through which genes obtain new functions, enabling one gene to maintain its original function and another to change its function [40]. Gene duplications are also considered one of the main forces of evolution and expansion of gene families [41]. Previous studies of *XTH*-family genes have shown that gene-fragment-repetition events have occurred in tobacco [19], soybeans [30], and other plants. Gene-duplication phenomena also occurred in the *BnXTH* gene family of ramie, with fragment-repetition events of the three pairs of isogenous genes (*BnXTH6*–*BnXTH16*, *BnXTH8*–*BnXTH9*, and *BnXTH17*–*BnXTH18*) (Figure 4a). This indicates that gene replication plays a certain role in evolution of *BnXTH* genes but is not the main driver of the expansion of the *BnXTH* gene family. The sequence similarity between the *BnXTH8*–*BnXTH9* and *BnXTH17*–*BnXTH18* gene pairs was significantly high (Figure 3a), and these two gene pairs exhibited similar expression patterns under Cd stress (Figure 6). Although the *BnXTH6*–*BnXTH16* gene pair also had high sequence similarity, its expression patterns were completely different under Cd stress. Similar observations were reported in *B. rapa*, where the *BraA.XTH32.a*–*BraA.XTH32.c* pair and the *BraA.XTH14.a*–*BraA.XTH14.b* pair showed similar expression patterns, while *BraA.XTH23.a* and *BraA.XTH23.b* showed different expression patterns [22]. This may have been because duplicate genes underwent nonfunctionalization, neofunctionalization, or subfunctionalization during evolution, resulting in similar or different gene-expression patterns [42]. There were few collinear gene pairs between ramie and monocotyledonous plants (*O. sativa* and *S. bicolor*) compared to the gene pairs detected between ramie and dicotyledonous plants (*A. thaliana* and *P. trichocarpa*). This indicates that numerous *XTH* gene variations and replications may occur in dicotyledonous plants during evolution. This evolution phenomenon was also found in the CAX-family genes of *P. trichocarpa* [43].

### 3.3. BnXTH Gene Response to Cd Stress

Abiotic stress can lead to transcript-level changes in *XTH* genes. For example, expression of *AtXTH14* and *AtXTH15* decreased significantly under Al stress, resulting in reduced XET activity and thus enhancing the Al tolerance of *A. thaliana* [26]. Additionally, expression of *PeXTH* was significantly upregulated in the roots and leaves of *P. euphratica* under Cd stress [44]. The current study found that the *BnXTH* gene family responded to Cd stress, under which *BnXTH1*, *BnXTH3*, *BnXTH6*, and *BnXTH15* were upregulated, while *BnXTH5*, *BnXTH16*, *BnXTH17*, and *BnXTH18* were significantly downregulated. Similar contrasting expression patterns of this gene family in response to abiotic stress have been reported in other plants. For example, expression of *CsXTH1*, *CsXTH4*, *CsXTH6*, and *CsXTH7* was upregulated, while that of *CsXTH3* was downregulated, in *Camellia sinensis* under fluorine stress [33]. Furthermore, we also found that BnXTH-family proteins have different subcellular localizations; for example, most BnXTH-family proteins were located in the cell wall, while BnXTH13 and BnXTH17 were located in the extracellular region. This may have been due to the expression-pattern diversity of the XTH gene family [18]. The differences in the subcellular localizations and expressions of proteins in the same family lead to differences in gene function [45], indicating that the different members of the *BnXTH* gene family exhibit different functions.

Expression of a gene often depends on the regulation of its upstream promoter [46]; thus, it is particularly important to analyze the upstream promoter sequences of a gene. The sequence analysis of the upstream promoter sequence of the *BnXTH*-family gene showed that the promoter of the *BnXTH*-family gene contained several cisacting elements, such as MYB, ABRE, AS-1, STRE, and MBS, that were involved in biotic and abiotic stress responses. MYB, MBS, and other ciselements are the binding sites of MYB transcription factors, which regulate defense responses by binding to the MBS elements on target genes [47]. Some hormone response elements, such as EREs and ABREs, are also involved in biotic and abiotic stress responses, whereby ABREs play important roles in response to abiotic stress [48,49]. These elements ensure that *BnXTH* genes are rapidly induced under stressful conditions.

Excessive absorption of heavy metals by plants causes serious toxicity to those plants [50]. Cell walls, especially hemicellulose, are reportedly the key Cd storage areas in plants [12]. In *A. thaliana*, phosphorus-deficiency tolerance significantly reduced hemicellulose content in the cell wall and alleviated Cd toxicity [51]. Heterologous expression of *PeXTH* in tobacco increased the root length and fresh weight of transgenic plants by enhancing their tolerance of Cd [44]. Similarly, in this study, the tolerance analysis of the transgenic yeast showed that heterologous expression of *BnXTH1*, *BnXTH6*, and *BnXTH15* under Cd stress could enhance the Cd tolerance of yeast cells (Figure 7). These results suggest that the *BnXTH* gene family is involved in Cd stress responses.

## 4. Materials and Methods

### 4.1. Identification and Analysis of BnXTH-Family Genes in the Ramie Genome

There are two conserved domains in XTH proteins: the Glyco_hydro_16 domain (PF00722) and the XET_C domain (PF06955) [22]. We generated a hidden Markov model (HMM) file of these two conserved domains using the Pfam database (https://Pfam.xfam.org/ (accessed on 13 October 2021)) [52]. The ramie genome was analyzed using HMMER v3.3.2 (Howard Hughes Medical Institute, Washington, DC, USA) [53], which identified candidate genes. Redundant sequences were manually removed, and the ramie *BnXTH* genes were finally obtained.

The corresponding *BnXTH* gene locations were obtained from the annotation file of the ramie genome and visualized on chromosomes via MG2C v2.1 online software (http://mg2c.iask.in/mg2c_v2.1/ (accessed on 20 October 2021)) [54]. ExPASy software (https://web.expasy.org/protparam/ (accessed on 20 October 2021)) [55] was used to predict the physical and chemical properties of the selected BnXTH-family members. These properties included each gene’s PI, Mw, GRAVY, and aliphatic index. Furthermore, Plant-mPLoc (http://www.csbio.sjtu.edu.cn/bioinf/plant-multi/ (accessed on 20 September 2022)) [56] was used to predict the subcellular localization of the BnXTH-family proteins.

### 4.2. Sequence Alignment and Phylogenetic Analyses

The XTH protein sequence of *A. thaliana* was searched for in the NCBI protein database (http://www.ncbi.nlm.nih.gov/protein/ (accessed on 21 September 2022)), and coding sequences (CDSs) of *BnXTH*-family genes were used to generate BnXTH protein sequences. A neighbor-joining (NJ) phylogenetic tree based on full-length sequences of AtXTHs and BnXTHs was constructed via MEGA 6.0 [57] using 1000 bootstrap replicates. A multiple sequence alignment of all BnXTH proteins was then conducted with Clustal (version:X 2.0, University College Dublin, Dublin, Ireland) [58].

### 4.3. Gene Structure and Motif Composition Analysis

The genomic sequences and CDSs of the *BnXTH*-family genes were extracted from the ramie genome, and structures of the *BnXTH*-family genes were constructed using GSDS 2.0 online software (http://gsds.gao-lab.org/ (accessed on 22 September 2022)) [59]. MEME online software (https://meme-suite.org/meme/tools/meme (accessed on 22 September 2022)) [60] was then used to analyze the conserved motifs of the BnXTH proteins, after which the conserved sites were set at 6–50 and the conserved-motif number parameter was set to 10. These structures were then visualized using TBtools (version: v1.098774, South China Agricultural University, Guangzhou, China) [61].

### 4.4. Analysis of Gene Duplication Events and Collinearity of the BnXTHs

We used a multiple-collinearity scanning toolkit (MCScanX) and TBtools software plug-ins to analyze gene duplication events of *BnXTH* in the ramie genome and the collinearity of the *XTH* genes between *Boehmeria nivea* and *O. sativa*, *Sorghum bicolor*, *A. thaliana*, and *Populus trichocarpa*. A collinearity graph was then generated using TBtools.

### 4.5. Analysis of the BnXTH Gene Promoter

The 2000 bp sequence upstream of the *BnXTH* gene was predicted using the PlantCARE database [62], and its cisregulatory elements were analyzed. These cisregulatory elements were then counted and classified according to the functional effects on the promoter. Thereafter, TBtools was used to visualize results and generate heat maps.

### 4.6. RT-qPCR Analysis of BnXTH Expression under Cd Stress

The “Xiangzhu No. 3” ramie material used in this study was provided by our research group at Hunan Agricultural University. Terminal buds were cultured in a half-strength Hoagland nutrient solution for 3 weeks, after which the roots of ramie seedlings with the same growth rate were collected. After 1 week of culturing in the half-strength Hoagland nutrient solution, the ramie seedlings were treated with 50 μM of CdCl_2_, and the root samples were collected at 0, 3, 6, 9, 12, 24, and 48 h after the treatment. The samples were frozen in liquid nitrogen and preserved at −80 °C. The growth conditions of the ramie were as follows: 14 h day/10 h night photoperiod, day and night temperature of 26/24 °C, relative humidity of 60%, and light intensity of 20,000 lux.

Total RNA was extracted according to the instructions of the plant RNA extraction kit (Vazyme, Nanjing, China). The extracted RNA was reverse-transcribed into cDNA using a reverse transcription kit (Vazyme, Nanjing, China) with primers that were designed using Premier 5.0 and synthesized via Sangon Biotech (Shanghai, China). The primer sequences are presented in Appendix A. The synthesized cDNA was used as a template for qPCR analysis on the Bio-Rad CFX96 instrument (Bio-Rad, Hercules, CA, USA) using the AceQ^®^ Universal SYBR qPCR kit (Vazyme, Nanjing, China) and the *BnActin* gene as the control. Each sample was quantified in triplicate, and the relative quantitation of each gene was conducted using the 2^−ΔΔCt^ method [21].

### 4.7. Functional Analysis of Cd-Induced Expression of BnXTHs in Yeast

The *BnXTH1*, *BnXTH3*, *BnXTH6*, and *BnXTH15* genes, which were shown to respond to Cd stress, were introduced into the yeast cells for functional analysis. Briefly, the ramie cDNA was used as a template for PCR, using the primers shown in Appendix A. The PCR conditions and procedures were as described by Jiang et al. [14]. Thereafter, the PCR products were cloned into the pEASY-blunt vector (TransGen Biotech Company, Beijing, China) and sequenced at Sangon Biotech (Shanghai, China), followed by subsequent cloning into the p426 GPD vector at the SmaI/SalI sites. The p426–*BnXTH1*/*3*/*6*/15 recombinant vector and the empty p426 vector were introduced into a mutant *Saccharomyces cerevisiae* yeast strain, *Δyap1*, which lacked transcriptional regulatory protein YAP-1 for Cd tolerance. For the Cd-tolerance analysis, the transgenic yeast cells were inoculated into a liquid synthetic dropout medium without uracil (SD-URA) and incubated at 30 °C on a shaker at 200 rpm until OD_600_ = 1.0 was reached. The precipitate was collected via centrifugation at 10,000 rpm for 1min, followed by suspension in ddH_2_O. The suspension was diluted to 10^−1^, 10^−2^, and 10^−3^ times its original state, and 2 µL of the 10^−2^ dilution was cultured as droplets on solid SD-URA media that contained 0 and 75 μM of CdCl_2_. The plates were incubated at 30 °C for 3 days, and the cultures were observed and photographed.

### 4.8. Statistical Analysis

All data are presented as means ± SD. The data were analyzed with one-way analysis of variance (ANOVA), followed by an LSD post hoc test using SAS 9.4 (SAS Institute, Cary, NC, USA) at a *p* ≤ 0.05 significance level.

## 5. Conclusions

This study identified 19 *BnXTHs* and evaluated their evolution, phylogeny, chromosomal locations, gene duplications, and cisregulatory elements. RT-qPCR results showed that most *BnXTH* genes responded to Cd stress. Many cisregulatory elements in *BnXTH* gene promoters were related to abiotic and biotic stress responses. In summary, under Cd stress, transcription factors that are located upstream of *BnXTHs* are triggered to bind to the cisregulatory elements that are upstream of *BnXTHs* and regulate expression of *BnXTH* genes to enhance the Cd tolerance of ramie. Additionally, functional analysis of heterologous expression in yeast showed that *BnXTH1*, *BnXTH6*, and *BnXTH15* may be involved in Cd tolerance. However, further validation, using transgenic methods, in ramie or other plants is needed. These results improved our understanding of the *BnXTH* gene family and laid a foundation for exploration of the function of *BnXTH* genes in Cd tolerance and enrichment in ramie.

## Figures and Tables

**Figure 1 ijms-23-16104-f001:**
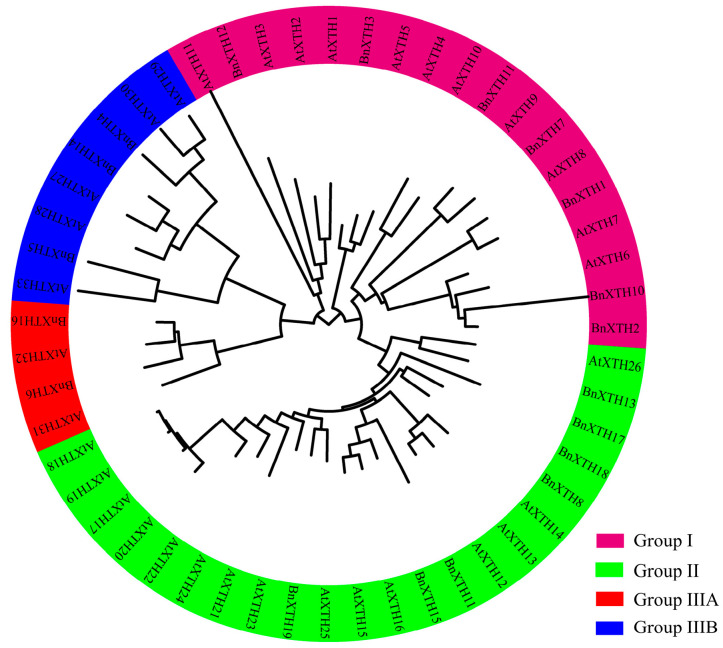
Phylogenetic tree showing the relationships among XTH proteins of *Boehmeria nivea* and those of *Arabidopsis thaliana*. The colored arcs show Groups I, II, IIIA, and IIIB. This phylogenetic tree was constructed via the neighbor-joining (NJ) method, using MEGA 6.06 with 1000 bootstrap replicates.

**Figure 2 ijms-23-16104-f002:**
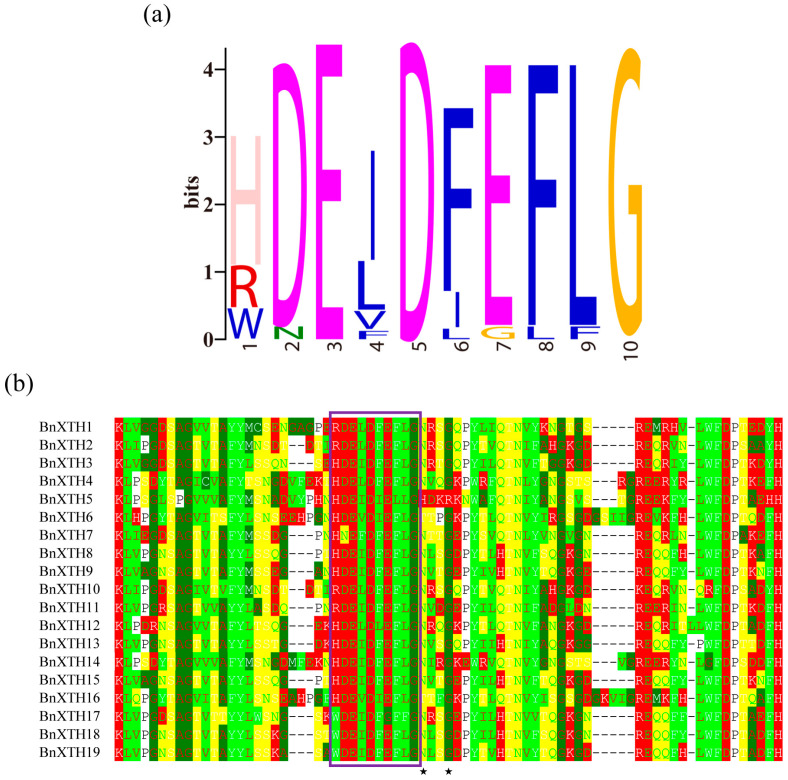
Logo diagram (**a**) and multiple sequence alignments (**b**) of the active catalytic regions of BnXTHs. The purple rectangular frames represent active catalytic regions (HDEIDFEFLG), and the asterisks represent N-linked glycosylation sites.

**Figure 3 ijms-23-16104-f003:**
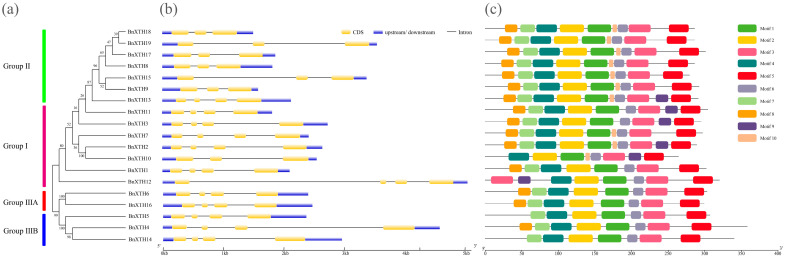
Phylogenetic (**a**), gene-structure (**b**), and conserved motif (**c**) analyses of the *BnXTH* gene family.

**Figure 4 ijms-23-16104-f004:**
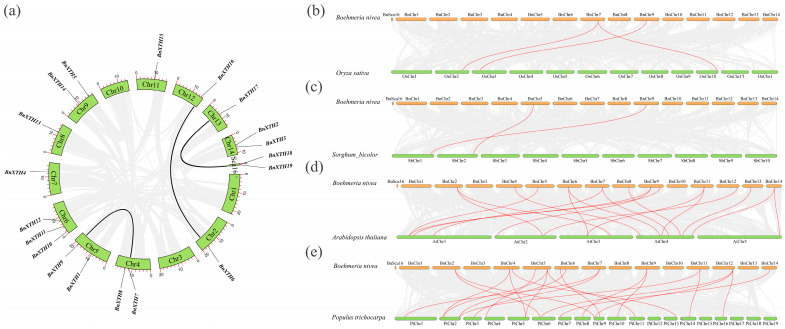
Chromosomal-level analyses of the *BnXTH* gene family in the ramie genome. (**a**) Synteny analysis of *BnXTHs.* The black lines indicate collinearity blocks and fragment-doubling events. (**b**–**e**) Synteny analysis of *BnXTHs* between *Boehmeria nivea* and *Oryza sativa*, *Sorghum bicolor*, *Arabidopsis thaliana*, and *Populus trichocarpa*.

**Figure 5 ijms-23-16104-f005:**
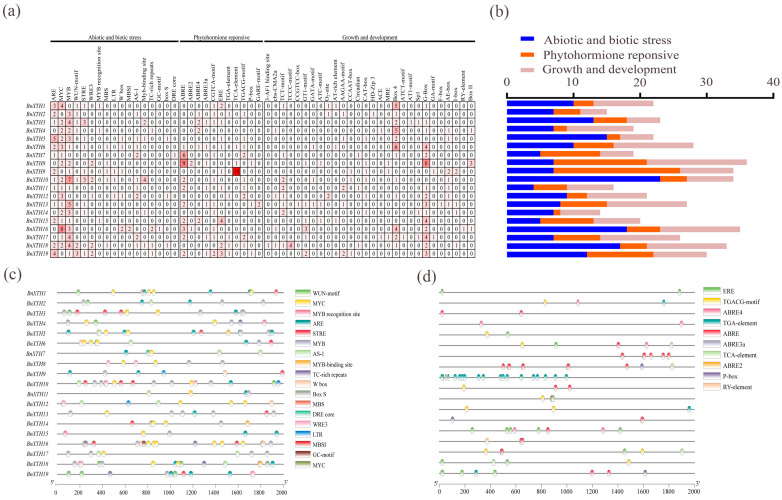
Prediction of cisregulatory elements of *BnXTHs*. (**a**) The number of cisregulatory elements in *BnXTH* promoters. (**b**) The number of cisregulatory elements involved in the responses to abiotic and biotic stress, phytohormone signaling, and plant growth and development. (**c**) Type, quantity, and position of the ciselements of *BnXTH* promoters in response to biotic and abiotic stress. (**d**) Type, quantity, and position of the ciselements of the *BnXTH* promoters in response to phytohormone signaling.

**Figure 6 ijms-23-16104-f006:**
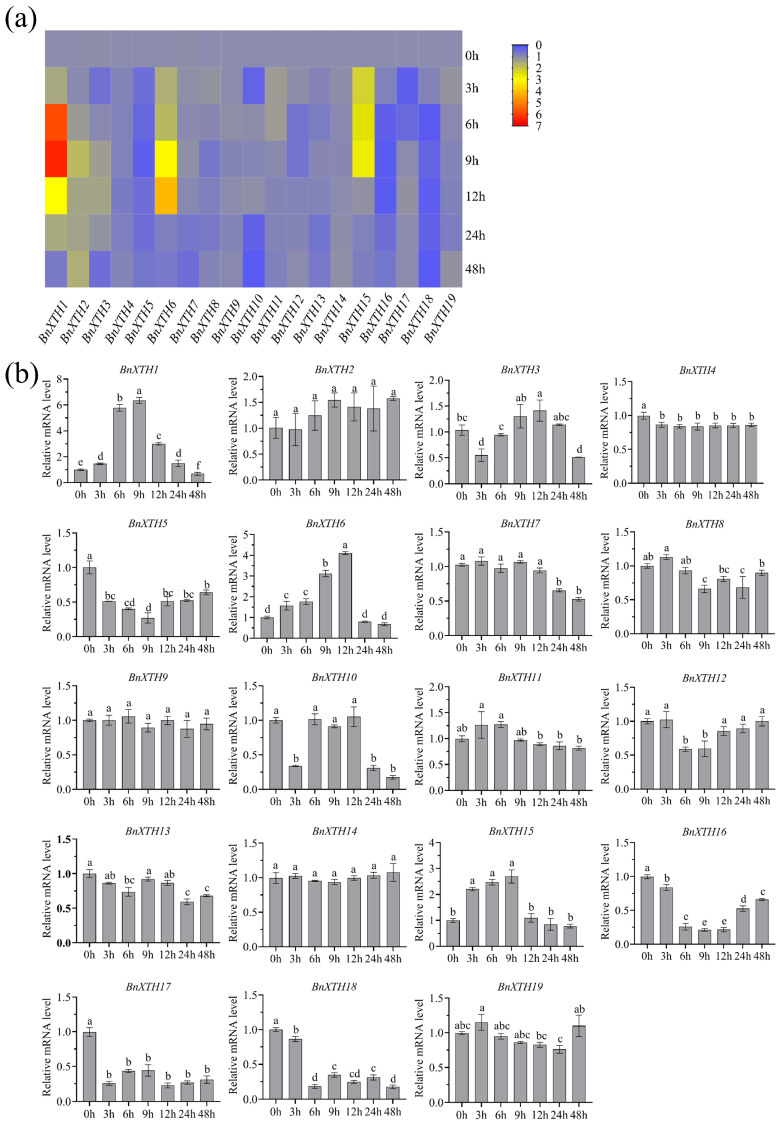
Expression analysis of the *BnXTH* genes under Cd stress. (**a**) Heat map of *BnXTH*-family genes expression level. (**b**) Expression of 19 *BnXTH* genes by qRT-PCR. Ramie seedlings were treated with Cd, and the root samples were collected at 0, 3, 6, 9, 12, 24, and 48 h after treatment. *Boehmeria nivea actin* (*BnActin*) was used as the internal control. Data are presented as means ± SD (*n* = 3). Statistical significance was determined using an LSD test. Different letters indicate significant differences at *p* ≤ 0.05.

**Figure 7 ijms-23-16104-f007:**
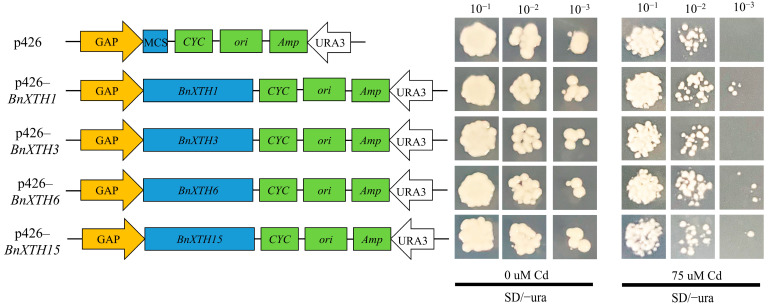
Functional analysis of *BnXTH* genes through heterologous expression in yeast. (**a**) Schematic diagram of the constructed yeast vector. p426 represents the empty vector, while p426–*BnXTH1*, p426–*BnXTH3*, p426–*BnXTH6*, and p426–*BnXTH15* represent the recombinant p426 vectors that contained the *BnXTH1*, *BnXTH3*, *BnXTH6*, and *BnXTH15* genes, respectively. (**b**) Cd-tolerance assay of *BnXTH* genes in transgenic yeast (*Δyap1*). The yeast cells were treated with 0 μM and 75 μM of Cd in an SD-URA medium for 3 days at 30 °C.

**Table 1 ijms-23-16104-t001:** Molecular characterization of *BnXTH* genes.

Name	Gene Name	Genome Location	PI	Mw (kDa)	Peptide Residue (aa)	GRAVY	Aliphatic Index	CDS Length (bp)	Predicted Subcellular Localization
*BnXTH1*	Bni05G007034	Chr5: 9195318–9197779	4.83	34.757	302	−0.536	62.02	909	Cell Wall
*BnXTH2*	Bni14G018696	Chr14: 919549–5922548	7.60	33.076	289	−0.362	70.55	870	Cell Wall
*BnXTH3*	Bni14G018759	Chr14: 6848323–6851051	8.59	33.88	395	−0.406	69.73	1188	Cell Wall/Cytoplasm
*BnXTH4*	Bni07G010864	Chr7: 13012796–13017451	8.57	40.228	358	−0.259	71.42	1077	Cell Wall
*BnXTH5*	Bni09G013633	Chr9: 15698329–15700698	7.16	34.447	307	−0.223	71.47	924	Cell Wall
*BnXTH6*	Bni02G003149	Chr2: 16541099–16543546	6.59	34.170	303	−0.559	61.82	912	Cell Wall
*BnXTH7*	Bni04G005825	Chr4: 11902190–11926110	8.77	32.334	286	−0.385	73.39	861	Cell Wall/Cytoplasm
*BnXTH8*	Bni04G006001	Chr4: 13734274–13737675	5.61	33.550	297	−0.315	67.27	894	Cell Wall
*BnXTH9*	Bni05G007965	Chr5: 18140865–18142082	9.47	32.9	292	−0.404	65.24	879	Cell Wall/Cytoplasm
*BnXTH10*	Bni06G008340	Chr6: 1295993–1298134	8.98	30.812	264	−0.548	67.58	795	Cell Wall
*BnXTH11*	Bni06G008558	Chr6: 4531957–4533699	6.46	35.716	304	−0.509	73.39	915	Cell Wall
*BnXTH12*	Bni06G008923	Chr6: 8233006–8239486	5.74	37.169	320	−0.800	59.44	963	Cell Wall
*BnXTH13*	Bni08G012131	Chr8: 12169484–12170900	9.11	33.171	291	−0.355	73.02	876	Extracell
*BnXTH14*	Bni09G013009	Chr9: 8910982–8915126	8.20	38.519	340	−0.478	69.12	1023	Cell Wall
*BnXTH15*	Bni11G015460	Chr11: 9722688–9725914	9.11	31.746	279	−0.323	71.58	840	Cell Wall/Cytoplasm
*BnXTH16*	Bni12G017196	Chr12: 15189643–15192043	9.31	34.254	299	−0.396	60.40	900	Cell Wall
*BnXTH17*	Bni13G017592	Chr13: 6362009–6373079	4.64	33.589	301	−0.347	65.42	906	Extracell
*BnXTH18*	BniUnG019321	Sca16: 619625–621073	8.69	31.917	286	−0.338	69.58	861	Cell Wall/Cytoplasm
*BnXTH19*	BniUnG019322	Sca16: 639410–654270	6.44	32.448	286	−0.329	70.94	861	Cell Wall/Cytoplasm

PI, isoelectric point; Mw, molecular weight; aa, amino acid; GRAVY, grand average of hydropathicity.

## Data Availability

Not applicable.

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
