# Peer review of "Identification of the Xyloglucan Endotransglycosylase/Hydrolase (XTH) Gene Family Members Expressed in Boehmeria nivea in Response to Cadmium Stress"

_ijms, 2022, doi:10.3390/ijms232416104_

Round 1

Reviewer 1 Report

I have gone through the manuscript entitled” Identification of the xyloglucan endotransglycosylase/hydrolases (XTH) gene family members expressed in Boehmeria niveain response to Cadmium stress” . The manuscript is related to the role of  Boehmeria nivea (ramie) XTH genes identified from Boehmeria nivea genome sequence and its response to Cadmium stress tolerance.In the manuscript the authors have selected some important XTH genes from genome sequence of ramie and they have functionally validated in these selected genes for cadmium tolerance via RT-PCR. Further they have checked their expression at transgenic level. This is an important study. However the figure 6 are not clearly visible. I advised the authors to include high resolution figure for the same.

Author Response

Dear Reviewer:

Thank you very much for your attention and the referee's evaluation and comments on our paper ijms-2090158. We have revised the manuscript according to your detailed suggestions. Enclosed please find the attachment.

Thank you very much for all your help and looking forward to hearing from you soon.

Sincerely yours,

Yushen Ma

Reviewer 2 Report

Add some data in the abstract

There are some linguistic mistakes in the abstract, check and throughout the manuscript

Abbreviation in Table 1 must be indicated in footnote

Figure 6 should be divided into two sub figure 6A and 6B for clarity

Clear in detail the mechanism of BnXTH gene in mitigating Cd on the level of protein content, antioxidant defense system etc,

Provide all devices origin and model

Line 411, Use LSD or Duncan as post hoc

Enhance conclusion

Update references, check the outputs of all references and consider the reference style regarding the year should be bold

Author Response

(The authors gave the same response as above.)

Round 2

Reviewer 2 Report

After checking the author's response, I recommended that this work can be accepted in the IJMS